

# COVID-19 onset reduced the sex ratio at birth in South Africa

Gwinyai Masukume[1], Margaret Ryan[2], Rumbidzai Masukume[3], Dorota Zammit[4], Victor Grech[5] and Witness Mapanga[6,7]

[1] Independent Researcher, Johannesburg, South Africa
[2] School of Social Work and Social Policy, Trinity College Dublin, Dublin, Ireland
[3] Department of Obstetrics and Gynaecology, Faculty of Health Sciences, University of the Witwatersrand, Johannesburg, South Africa
[4] National Statistics Office, Valletta, Malta
[5] Academic Department of Paediatrics, Medical School, Mater Dei Hospital, Msida, Malta
[6] Division of Medical Oncology, Department of Medicine, School of Clinical Medicine, Faculty of Health Sciences, University of the Witwatersrand, Johannesburg, South Africa
[7] Noncommunicable Diseases Research Division, Wits Health Consortium (PTY) Ltd., Johannesburg, South Africa

Corresponding author
Gwinyai Masukume,
parturitions@gmail.com

## ABSTRACT

**Background.** The sex ratio at birth (defined as male/(male+female) live births) is anticipated to approximate 0.510 with a slight male excess. This ratio has been observed to decrease transiently around 3–5 months following sudden unexpected stressful events. We hypothesised that stress engendered by the onset of the COVID-19 pandemic may have caused such a decrease in South Africa 3–5 months after March 2020 since in this month, South Africa reported its first COVID-19 case, death and nationwide lockdown restrictions were instituted.

**Methods .** We used publicly available, recorded monthly live birth data from Statistics South Africa. The most recent month for which data was available publicly was December 2020. We analysed live births for a 100-month period from September 2012 to December 2020, taking seasonality into account. Chi-squared tests were applied.

**Results.** Over this 100-month period, there were 8,151,364 live births. The lowest recorded monthly sex ratio at birth of 0.499 was in June 2020, 3 months after March 2020. This June was the only month during this period where the sex ratio inverted *i.e.*, fewer male live births occurred. The predicted June 2020 ratio was 0.504. The observed June 2020 decrease was statistically significant $p = 0.045$.

**Conclusions.** The sex ratio at birth decreased and inverted in South Africa in June 2020, for the first time, during the most recent 100-month period. This decline occurred 3 months after the March 2020 onset of COVID-19 in South Africa. As June 2020 is within the critical window when population stressors are known to impact the sex ratio at birth, these findings suggest that the onset of the COVID-19 pandemic engendered population stress with notable effects on pregnancy and public health in South Africa. These findings have implications for future pandemic preparedness and social policy.

## INTRODUCTION

The sex ratio at birth (SRB) also known as the secondary sex ratio is defined as (male/(male + female) live births) (*Grech, 2014*). At the population level, it approximates 0.510 with male live births slightly exceeding female live births. The SRB may act as a sentinel health indicator underscoring adverse circumstances through its decrease, but also highlighting improved circumstances with its increase (*Davis, Gottlieb & Stampnitzky, 1998*; *Grech & Masukume, 2016*). Following sudden unexpected stressful events, the population SRB has been observed to decrease transiently usually some 3–5 months following events such as the death of a highly popular public figure (*Grech, 2015a*), terrorist attacks (*Masukume et al., 2017*) and wide spread protesting/rioting (*Grech, 2015b*). Such stressful life events impact on maternal neuroimmunoendocrine condition; male fetuses are more vulnerable to such changes in condition (*Aiken & Ozanne, 2013*; *James & Grech, 2017*). Another potential window is 9 months after such events as was the case in Japan when the SRB dipped 9 months after the Kobe earthquake (*Fukuda et al., 1998*). However, these windows of SRB decline are not a universal phenomenon due to varying contextual factors (*Bruckner et al., 2019*; *Grech & Scherb, 2021*).

It has been proposed that COVID-19 stress might result in SRB decline (*Abdoli, 2020*). Indeed in Japan, a decrease in SRB was seen in December 2020, 9 months after the COVID-19 pandemic was declared (*Inoue & Mizoue, 2022*). In March 2020, South Africa reported its first known polymerase chain reaction (PCR) confirmed COVID-19 case (*Jassat et al., 2021*). During that same month the World Health Organization declared COVID-19 to be a pandemic (*Cucinotta & Vanelli, 2020*), and this was accompanied by a surge in local and global media attention on COVID-19 (*Ng, Chow & Yang, 2021*). In addition, during March 2020, South Africa declared a national state of disaster and instituted nationwide lockdown measures which included school closures, and restrictions on gatherings, domestic and international travel (*Abdool Karim, 2020*). Furthermore, the first death attributed to COVID-19, in South Africa also occurred (*Pillay-van Wyk et al., 2020*). In short, March 2020 was a momentous month in relation to COVID-19 in South Africa.

Our primary hypothesis was that stress engendered by the onset of the COVID-19 pandemic might have caused a transient decrease in SRB in South Africa 3–5 months after March 2020. The secondary hypothesis was that an SRB dip occurred 9 months later, in December 2020.

## MATERIAL AND METHODS

### Data and statistical analysis

Monthly male and female recorded live births, from the national birth registration system, were obtained from Statistics South Africa's publicly available annual reports (*Statistics South Africa, 2021*). For the intercensal period 2011–2016, the completeness of birth registration was estimated at 88.6%. Data was extracted for a 100-month period from September 2012 to December 2020. The last year the SRB, in South Africa, was known to have been influenced by an event was in 2011 when it increased significantly 9 months

after the 2010 FIFA World Cup (*Masukume & Grech, 2015*). December 2020 was the most recent month for which data was available. We conducted a post-hoc analysis considering 3–5 and 9 months after the April 2015 protests/riots (*Desai, 2015*), as the SRB might have been affected consequent to stress induced by this event.

To take seasonality into account, similar months across the years were compared. Restriction (*Howards, 2018*) was used to control for potential confounding factors. For parsimony, chi-squared tests were utilized (*Gauch Jr, 2002*). This methodology is an alternative to time series analysis that has previously been used with similar South African data (*Masukume, Grech & Scherb, 2016*). Statistical significance was defined as a two-tailed *p*-value < 0.05. For analysis, Stata version 17BE (College Station, TX, USA) was used.

### Ethical considerations
Due to the anonymized nature of the data, ethical approval was not required.

## RESULTS

Over this 100-month period, from September 2012 to December 2020, there were a total of 8,151,364 live births as 4,113,288 males and 4,038,076 females, SRB 0.505. The lowest recorded monthly sex ratio at birth, during the 100-month period, was in June 2020 at 0.499 (Fig. 1). June 2020 was also the only month during this most recent 100-month span that the SRB inverted *i.e.*, fewer male than female live births occurred. In addition, Fig. 1 showed that the July and August 2020 SRBs of 0.503 and 0.504, respectively, the fourth and fifth months after the COVID-19 pandemic began in South Africa, were consistent with preceding years for those months. The SRB for December 2020 was 0.503. Figure 1 was also used to perform a visual assessment of SRB seasonality, which found that it was limited. The SRB for a specific year was predicted using a two-year moving average from previous years because seasonality was limited and restriction was used to control for confounding. Based on similar months in previous years, over the 100-month period, the moving average approach predicted an SRB of 0.504 for June 2020 (Figs. 2 and 3) and 0.505 for December 2020 respectively. The June 2020 SRB decrease was statistically significant *p* = 0.045, while for December 2020 there was no statistically significant change *p* = 0.293 (Table 1). Post-hoc analysis, Table S1, revealed no statistically significant result.

## DISCUSSION

In this study, we provide insight into how the SRB, in South Africa, was influenced by the onset of the COVID-19 pandemic. Over the study period of 100 months, we observed the lowest SRB of 0.499 in June 2020, 3 months after the first COVID-19 case, death and lockdown occurred in March 2020. Notably, the SRB had inverted *i.e.*, fewer male live births occurred. This is the only month this happened during the 100-month period, from September 2012 through to December 2020. Our findings are inline with the hypothesis that COVID-19 stress would result in SRB decline (*Abdoli, 2020*), driven by the Trivers-Willard hypothesis which posits that adverse environmental conditions can lower the ratio of males to females (*Trivers & Willard, 1973*). Our findings are also consistent with studies that have reported a transient SRB decline 3–5 months following unexpected events which

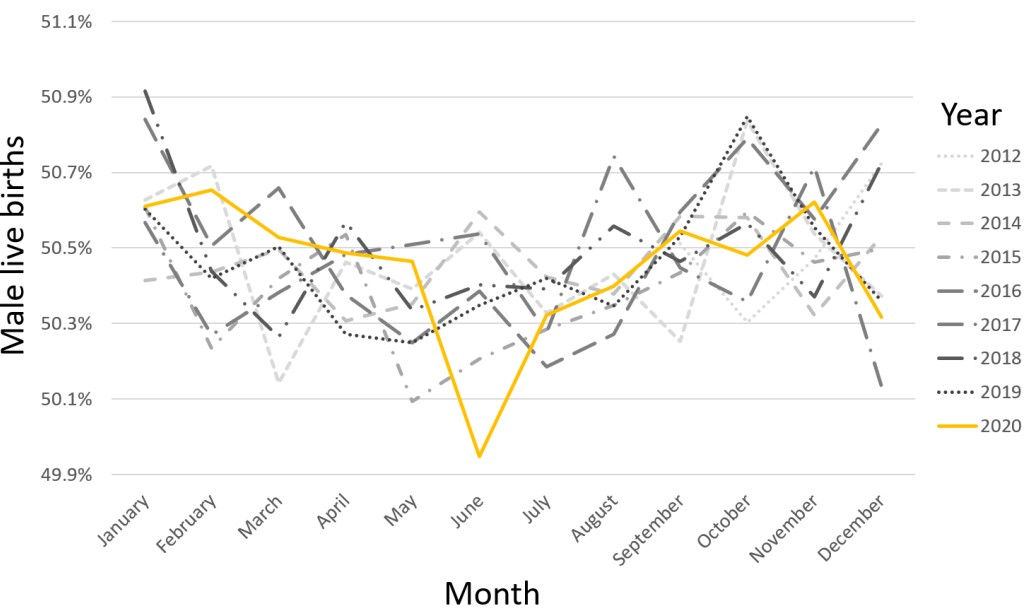

**Figure 1** Monthly proportion of male live births over 100 months from September 2012 to December 2020.

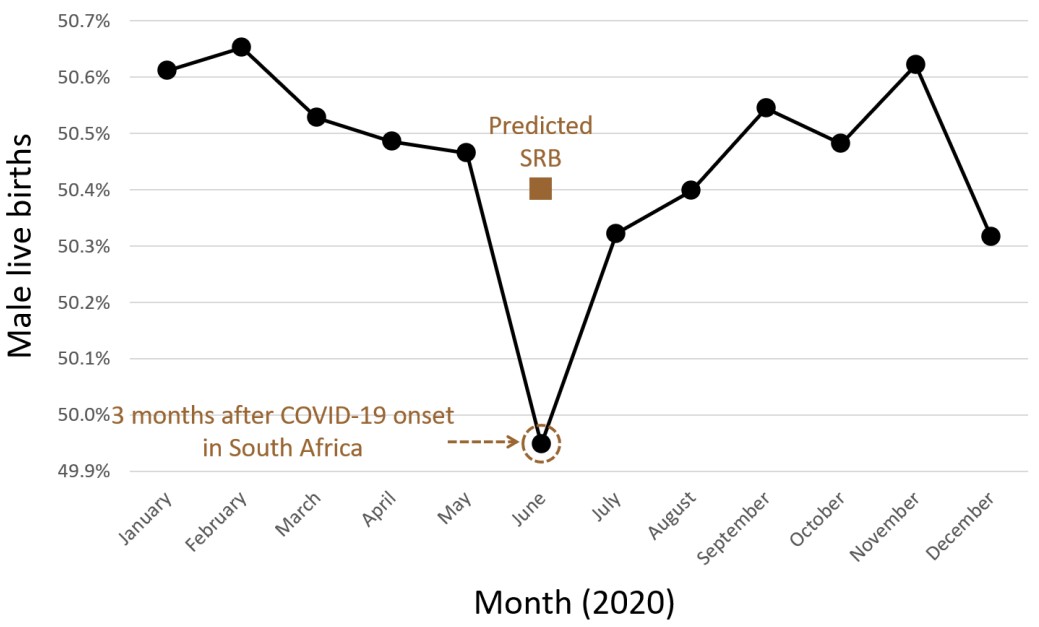

**Figure 2** Monthly proportion of male live births in 2020 with moving average.

stressed populations acutely (*Calleja, 2020*; *Grech, 2015a*; *Grech, 2015b*; *Masukume et al., 2017*; *Retnakaran & Ye, 2020*). A sex ratio at birth of less than 0.5 has also been observed in the 3–5 months following the July 2011 Norway attacks, the December 2012 Sandy Hook Elementary School shooting, and the October 2017 assassination of investigative journalist

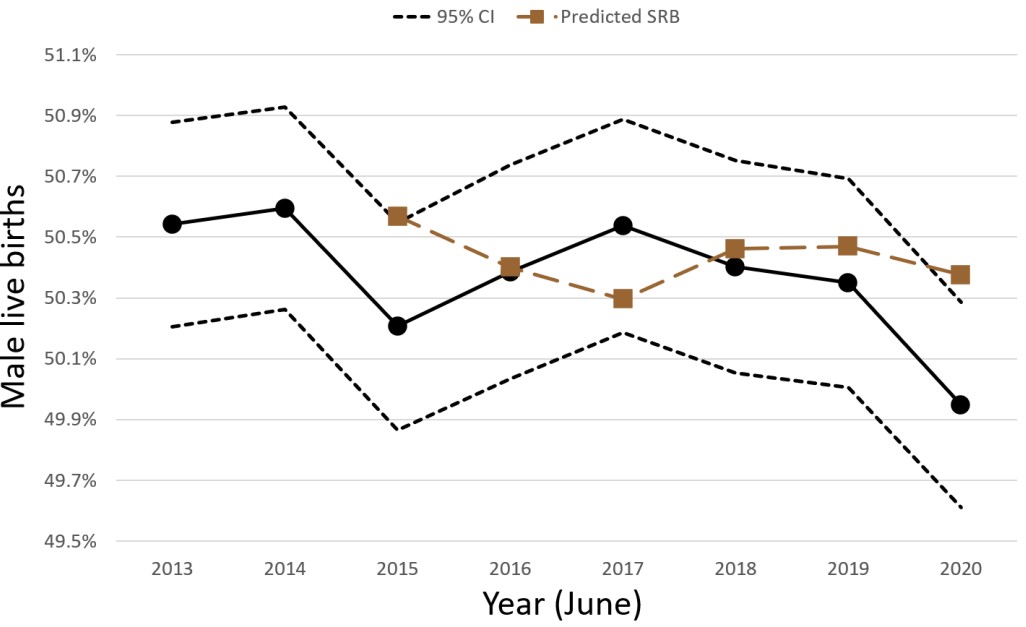

**Figure 3** Proportion of male live births during the month of June from 2013 to 2020.

**Table 1 Sex ratio at birth restricted from 2018 to 2020.**

| Period | Male live births | Female live births | Sex ratio at birth | 95% CI |
|---|---|---|---|---|
| June 2018 + 2019 | 81,078 | 79,870 | 0.504 | 0.501–0.506 |
| June 2020 | 42,022 | 42,109 | 0.499 | 0.496–0.503 |
| | Pearsons's Chi-squared = 4.0292 | | | |
| | $P = 0.045$ | | | |
| December 2018 + 2019 | 81,237 | 79,495 | 0.505 | 0.503–0.508 |
| December 2020 | 41,795 | 41,268 | 0.503 | 0.500–0.507 |
| | Pearsons's Chi-squared = 1.1058 | | | |
| | $P = 0.293$ | | | |

**Notes.**
CI, confidence interval.

Daphne Caruana Galizia (*Calleja, 2020*; *Grech, 2015b*). Although an SRB decline has been reported 9 months after an acute unanticipated severe stressor (*Fukuda et al., 1998*), we did not find evidence of this in South Africa, and the SRB, in December 2020, was not appreciably different in comparison to previous Decembers.

The timing of the SRB dip, 3 months following the onset of COVID-19 in South Africa suggests a mechanism operating in ongoing pregnancies whereby, due to stress, there is an elevated risk of fetal death among males (*Catalano et al., 2005*) resulting in fewer live born males. Thus beyond the direct adverse effects of the pandemic pathogen on pregnant women evidenced by a higher maternal mortality rate among women admitted with SARS-CoV-2 infection (*Budhram et al., 2021*), anxiety and stress likely generated

by news of the impending pandemic (*Ng, Chow & Yang, 2021*), in March 2020, led to disproportionate male fetal loss. Triangulation with antepartum deaths from South Africa's perinatal mortality audit system, when the data becomes publicly available, would be needed to directly confirm this excess male fetal loss (*Lavin et al., 2018*). Indeed, preliminary findings, though limited by missing data, show that the stillbirth rate in South Africa increased during the COVID-19 pandemic (*Pattinson et al., 2021*). Had a dip been witnessed in December 2020, 9 months after the onset of COVID-19, this would have pointed to a conception mechanism. As previously stated, a decrease in the SRB, as seen in this study, happens more frequently 3–5 months following a discrete population stressor. The different findings where the SRB fell in Japan 9 months after the COVID-19 pandemic was declared (*Inoue & Mizoue, 2022*) could be possibly explained by Japan's distinct population reaction to an acute stressor (*Fukuda et al., 2018*).

It is plausible that the June 2020 SRB decline could have been attributed to an event approximately 9 months earlier (*Fukuda et al., 1998*), in September 2019. Indeed there were protests/riots during that September which garnered significant media attention (*Tarisayi & Manik, 2020*). However, several factors are not consistent with this paradigm of the South African June 2020 SRB decrease being primarily related to September 2019 events. First, there is the precedent of the Los Angeles protests/riots (*Grech, 2015b*) where the SRB perturbation occurred during the 3–5 month post-event window. Second, similar protests/riots to the aforementioned ones have occurred before in South Africa during the most recent 100-month period, in April 2015 (*Desai, 2015*), but with no significant SRB decline either 3–5 or 9 months later (post-hoc analysis). Third, contrary to COVID-19, which had broader nationwide ramifications, these protests (*Desai, 2015*; *Grech, 2015b*; *Mavunga, 2019*) were confined to certain locations with possibly more localised effects.

Ecological fallacy, the limited ability to extrapolate findings to the individual level applies to this study (*Hart, 2011*). Nonetheless, the ecologic study technique is arguably best suited to investigate how a population outcome is influenced by a population stressor (*Pearce, 2011*). In late March 2020, due to the institution of nationwide lockdown measures, the Department of Home Affairs (DHA) only offered limited civic services. Due to this, there was a decline in the number of live births registered early (<30 days). With the easing of restrictions, services resumed in early May 2020. In 2020, there was an overall compensatory increase in the number of live births registered late (>30 days) (*Statistics South Africa, 2021*). The neonatal mortality rate, death within the first 28 completed days of life, in South Africa approximates 12 deaths per 1,000 live births (*Damian et al., 2019*). Nonetheless, the total recorded live births for 2020 is inline with prior years. Overall, this means when restrictive measures were lifted those who had missed registering a live birth came forward. Thus, the present analysis is unlikely to have been affected by closure of the DHA offices necessitated by lockdown regulations.

The present analysis covers the first COVID-19 wave in South Africa. At the time of writing, South Africa had experienced five waves (*Tegally et al., 2022*). It is possible that the SRB may have been transiently perturbed in the following waves. This is an area for research when the recorded live birth data for these periods become available.

## CONCLUSIONS

The sex ratio at birth decreased and inverted in South Africa in June 2020, for the first time, during the most recent 100-month period. This decline occurred 3 months after the March 2020 onset of COVID-19 in South Africa. As June 2020 is within the critical window when population stressors are known to impact the sex ratio at birth, via elevated risk of fetal death among males, these findings suggest that the onset of the COVID-19 pandemic engendered population stress with notable effects on pregnancy and public health in South Africa. Our findings can help inform future pandemic preparedness (*Marston, Paules & Fauci, 2017*) and social policy (*Cook & Ulriksen, 2021*). Examples of how this knowledge might be put to use to avert a possible SRB drop include targeted increased resourcing for maternal health services, including enhanced fetal surveillance, specifically during and in the lead up to 3–5 months after the onset of a future pandemic. Whether other countries experienced a decline in the SRB, following the onset of COVID-19 is an area for future research.

## ACKNOWLEDGEMENTS

We acknowledge Statistics South Africa for the recorded live birth data.

### Funding

The authors received no funding for this work.

### Competing Interests

The authors declare there are no competing interests.

### Author Contributions

- Gwinyai Masukume conceived and designed the experiments, performed the experiments, analyzed the data, prepared figures and/or tables, authored or reviewed drafts of the article, and approved the final draft.
- Margaret Ryan conceived and designed the experiments, performed the experiments, authored or reviewed drafts of the article, and approved the final draft.
- Rumbidzai Masukume conceived and designed the experiments, performed the experiments, authored or reviewed drafts of the article, and approved the final draft.
- Dorota Zammit conceived and designed the experiments, performed the experiments, analyzed the data, authored or reviewed drafts of the article, and approved the final draft.
- Victor Grech conceived and designed the experiments, performed the experiments, authored or reviewed drafts of the article, and approved the final draft.
- Witness Mapanga conceived and designed the experiments, performed the experiments, authored or reviewed drafts of the article, and approved the final draft.

## Human Ethics

The following information was supplied relating to ethical approvals (*i.e.*, approving body and any reference numbers):

Due to the anonymized nature of the data, ethical approval was not required.

## Data Availability

The data is publicly available at Statistics South Africa: https://www.statssa.gov.za/?page_id=1854&PPN=P0305.

## Supplemental Information

Supplemental information for this article can be found online at http://dx.doi.org/10.7717/peerj.13985#supplemental-information.

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
