# Peer review of "COVID-19 onset reduced the sex ratio at birth in South Africa"

_PeerJ, doi:10.7717/peerj.13985_

## Round 0.1 · original submission · Minor Revisions

Please pay attention to comments from both reviewers and make specific amendments.

Reviewer 1 ·

Basic reporting

There is no doubt that the COVID-19 pandemic has had an enormous impact on people worldwide, and women in pregnancy must be one of them, which is the manuscript's topic. The authors tried to explore such influences by focusing on the sex ratio at birth (SRB) in South Africa. The manuscript was well written, and the message of that was straightforward. However, there are a few comments that I would like the authors to consider.

Experimental design

no comment

Validity of the findings

SRBs were calculated referred to the observed data in South Africa from 2012 to 2020. It would benefit the readers if the authors could add the SRBs' uncertainties (95% CI) not only in the text but also in the table and figures.

Is it possible to explore the geographical heterogeneity of the SRB in South Africa? The impacts of the pandemic, for instance, national lockdown measures and school closures, must have varied between areas.

Additional comments

Please explain L109 more clearly; otherwise, the readers could not follow it.
With regard to figure 2 and 3, I think the moving averages should be drawn as lines rather than dots.

Reviewer 2 ·

Basic reporting

No comment

Experimental design

No comment

Validity of the findings

No comment

Additional comments

Thank you for giving me the opportunity to read this manuscript. Using publicly available demographic data in South Africa, the authors examined if sex ratio at birth (SRB) (defined as the proportion of male live birth to total live births) declinded after the coronavirus 2019 (COVID-19) pandemic. They found that SRB declined three months after the COVID-19 affected the country. While their statistical approach was more simple than those employed in the previous literature on the same topic, it seems that the manuscript presents robust (and interesting) results. I have some questions and suggestions for the authors.

(L.50-51) The authors concluded that their "findings have implications for future pandemic preparedness and social policy." I am not sure as to what they (or the South African government) can do practically based on their findings.

(L.99-100) "Our primary hypothesis was that...3-5 months after March 2020".
While they hypothesized that SRB declinded 3-5 months after March 2020, they only tested the difference in SRB observed in June 2020 and June 2018-2019, but not in relation to July or August. Please fix this inconsistency in the manuscript.

(L,131-132) Please describe in detail "the moving average approach", Please also provide the explanation in footnotes of Figures 2 and 3.

---

## Round 0.2 · accepted · Accept

The authors should be congratulated on their successful revisions.

Reviewer 1 ·

Basic reporting

no comment

Experimental design

no comment

Validity of the findings

no comment

Additional comments

no comment

Reviewer 2 ·

Basic reporting

no comment

Experimental design

no comment

Validity of the findings

no comment

Additional comments

The authors have successfully improved their manuscript. I have no additional comments.